# Hikikomori (Severe Social Withdrawal) in Italian Adolescents: Clinical Features and Follow-Up

**DOI:** 10.3390/children10101669

**Published:** 2023-10-09

**Authors:** Greta Tolomei, Gabriele Masi, Annarita Milone, Pamela Fantozzi, Valentina Viglione, Antonio Narzisi, Stefano Berloffa

**Affiliations:** IRCCS Stella Maris, Scientific Institute of Child Neurology and Psychiatry, Viale del Tirreno, 331A, Calambrone, 56128 Pisa, Italy; greta.tolomei@fsm.unipi.it (G.T.); annarita.milone@fsm.unipi.it (A.M.); pamela.fantozzi@fsm.unipi.it (P.F.); valentina.viglione@fsm.unipi.it (V.V.); antonio.narzisi@fsm.unipi.it (A.N.); stefano.berloffa@fsm.unipi.it (S.B.)

**Keywords:** hikikomori, social withdrawal, Internet gaming disorder, adolescence, behavioral addiction

## Abstract

Severe social withdrawal, including staying alone in one’s bedroom, non-attendance at school or work, and minimal or absent social contacts, sometimes only through electronic devices, can be found in several psychiatric disorders, or in a ‘primary’ form, firstly defined in Japan as ‘Hikikomori’. The distinction between primary and secondary forms is questionable, as it prevalently depends on the quality of psychiatric assessment. To date, few studies specifically explored Hikikomori in an adolescent population outside Japan. The aim of the present study is to describe clinical features of a consecutive group of 80 referred youth (13 to 18 years, 57 males) with social isolation, of which 40 were followed up on for 4–6 months, to characterize clinical features and outcome. All the participants presented psychiatric comorbid disorders, prevalently anxiety disorders, mood disorders, and autism spectrum disorder. Suicidality (ideation and behavior) was reported in 32.5% of the participants, and 20% of the participants attempted suicide. More than half of the participants exceeded the cut-off of the Internet Addiction Test, and 42.4% met the criteria for the Internet Gaming Disorder. At the follow-up appointment, an improvement of social withdrawal was reported in 75% of the sample; 67.5% of the participants significantly improved according to the CGI-improvement scale; and 55% of the participants had an improvement of functioning according to the C-GAS. Our findings suggest that Hikikomori is a transnosographic entity, with high rates of suicidality and Internet addiction, and that can it improve when it is timely diagnosed and treated.

## 1. Introduction

Severe social withdrawal and self-exclusion include non-participation in school or work, minimal social contacts or social contacts maintained only through electronic devices, and staying at home or secluded in one’s bedroom. There is a prototypical or ‘primary’ form of the condition, often defined as Hikikomori, which was first described in Japan, that has become a model for similar clinical presentations in different countries, with different cultural perceptions of the phenomenon [1,2] Social withdrawal can be a symptom of several psychiatric disorders such as schizophrenia, unipolar or bipolar depression, anxiety disorders (particularly generalized and social anxiety), autism spectrum disorder, some personality disorders (i.e., avoidant, schizotypal schizoid), and post-traumatic stress disorder (PSTD) [1,2,3]. In recent years, cases of Hikikomori have been reported in countries outside Japan [4,5] and, what appeared to be a phenomenon exclusively bonded to the Japanese culture, now seems to be a more global issue [4]. The male/female ratio is usually reported as unbalanced, with a heavier distribution among males. The onset of Hikikomori, or at least the prodromal signs, occurs usually during adolescence [3]. In a study by Koyama et al., 37.4% of Hikikomori cases had an onset of symptoms between 15 and 19 years, and 20% between 10 and 14 years [6]. Despite this finding, few studies in the literature are focused on these age ranges as almost all the Hikikomori studies include patients with a wide age range, with adolescents and adults in the same samples. We can assume that adolescents with Hikikomori may present specific features, treatment needs, and changes over time, and may represent a specific subtype of the disorder.

New international diagnostic criteria were developed in 2017 [7] to help diagnosis of the disorder, especially outside Japan: (1) marked social withdrawal, with seclusion in one’s own home; (2) continuous social isolation for at least six months; and (3) significant functional impairment or distress associated with social isolation. Individuals with a duration of continuous social withdrawal of at least three months should be considered pre-Hikikomori. Several specifiers have been proposed such as a lack of social participation, a lack of social interactions, experiences of loneliness, and, of course, a co-occurring psychiatric condition. These specifiers are helpful for a further characterization of Hikikomori, especially if it is aimed to the development of a rehabilitation project [8].

A clinically relevant distinction can be made between egodystonic or egosyntonic social withdrawal. In the egodystonic form, people wish to escape from the prison of their isolation but are unable to do so. In the egosyntonic form, isolation is felt as a ‘free choice’ to avoid a threatening or boring world, or to engage in interesting activities that are much more attractive than social interactions. Some individuals may present both egodystonic and egosyntonic characteristics, and these behaviors may change over time, as well as their thoughts and emotions regarding social relationships [3].

The distinction between primary and secondary forms is strongly influenced by the type of assessment used to diagnose comorbid psychiatric disorders. For example, most Hikikomori-like conditions, initially considered as the prototypical primary social withdrawal, are often associated with other mental disorders, incorrectly underdiagnosed or misdiagnosed, due to a lack of proper assessment [3,4,5,6]. Furthermore, the increasing diffusion of Internet use among youth has determined, in the last decade, a high rate of potentially addictive behaviors, composing the new category of Internet Gaming Disorder (IGD), often associated with severe social isolation [9]. Some symptoms such as anhedonia, fear of social judgement, and negative thoughts associated with stressful environmental conditions can induce pathological social withdrawal; likewise, social withdrawal behavior can in turn induce the same psychiatric symptoms. However, when social isolation is in comorbidity with other psychopathological disorders, these specific associations have prognostic and treatment implications [3]. Thus, identifying the primary disorder can be difficult. We propose that the distinction between a primary social withdrawal, corresponding to the Hikikomori, and a secondary social withdrawal, may be misleading, hampering the comparison of samples from different countries. The clinical picture of severe social withdrawal may be considered a specific transnosographic psychopathological feature, shared by patients from different categorical diagnoses, which may be defined as Hikikomori. If social withdrawal may be considered as a psychopathological dimension, with a continuum between normal and pathological extremes, Hikikomori should be considered as the most severe part of the continuum.

Our study aimed to determine the epidemiological and clinical characteristics of a group of 80 young adolescents with social withdrawal, and to distinguish possible clinical subtypes and developmental trajectories. Clinical data were analyzed at baseline and, in a subgroup of patients, at a 4–6-month follow-up, after a treatment program involving neuropsychiatric monitoring, rehabilitation treatment, and/or pharmacotherapy. The results were used to assess the clinical course and the developmental trajectories of the population with social withdrawal, and to identify which measures were more helpful to best monitor the course of these patients.

## 2. Materials and Methods

### 2.1. Sample

This was a naturalistic study based on a clinical database of referred children and adolescents, aged between 7 and 18 years, that had been consecutively referred as inpatients or outpatients to our third level university hospital with nation-wide catchment during a 3-year period (January 2020–January 2023). The inclusion criterion was the presence of social withdrawal, fitting all the following features: (1) spends most of the day and days at home; (2) avoids social situations, such as attending school or other activities; (3) avoids social relationships, such as friendships or contact with family members; (4) experiences significant distress or deterioration due to social isolation; and (5) occurs for a minimum duration of six months. These symptoms had to be a highly predominant clinical feature over the other co-occurring symptoms. All the subjects were screened for psychiatric disorders, using historical information and a structured clinical interview, the Schedule for Affective Disorders and Schizophrenia for School-Age Children-Present and Lifetime Version (K-SADS-PL) [10]. The prodromal period prior the first visit was retrospectively controlled for other psychiatric disorders and their time of onset, according to both parent and subject reports, as well as to chart reviews and direct contact with their previous treating psychiatrists. All the patients with schizophrenia, intellectual disability, and poor verbal skills (expression and/or comprehension) were excluded. All the participants had a cognitive functioning in the normal/borderline range (IQ > 70), assessed by standardized testing.

Based on the above-mentioned criteria, our study sample included 80 subjects with social withdrawal (57 males and 23 females, with an age range of 13.2–17.9 years and a mean age of 15.2 years ± 1.8 years). All the subjects underwent a comprehensive psychiatric assessment including medical history, standardized questionnaires, and clinical observations. A psychiatric follow-up was repeated in 40 subjects after 4–6 months from the initial assessment.

### 2.2. Procedures

All the participants received a clinical assessment using a clinical interview, the K-SADS-PL [10], to obtain a diagnosis based on the DSM-5 criteria. The age of onset of the social withdrawal symptomatology, a possible correlation with the COVID-19 pandemic lockdown, a presence of suicidal ideation and or/ previous suicide attempts, and ongoing pharmacological and/or psychotherapeutic treatments were specifically explored. The clinical severity and functional impairment were explored using the Clinical Global Impressions Severity Scale (CGI-S) [11] and the Children’s Global Assessment Scale (CGAS), respectively [12].

The Child Behavior Checklist 6-18 (CBCL 6-18) Parent Report Form was administered to the parents for a comprehensive dimensional assessment of psychopathology [13]. Two widely used measures were used for the assessment of anxiety: the Pediatric Anxiety Rating Scale (PARS), a clinician rated measure for anxiety [14], and the Multidimensional Anxiety Scale for Children 2nd Edition—Self-Report (MASC-2 SR), for a self-rated assessment of anxiety [15]. The Children’s Depression Inventory (CDI-2), a self-rated measure for depression [16], was used to explore depressive feelings associated with the subjects’ social withdrawal.

Two measures were used to explore the Hikikomori symptomatology: the Hikikomori Questionnaire (HQ-25), a self-rated assessment of Hikikomori symptoms [17], and the Tarumi’s Modern-Type Depression Trait Scale (TACS-22), a self-rated assessment of depressive features closely related to Hikikomori symptomatology [18]. The HQ-25 was recently validated in an Italian adult population, with good psychometric properties [19], while, for TACS-22, psychometric characteristics are not available in Italian samples.

Finally, 65 subjects completed the Internet Addiction Test (IAT) [20,21] and 59 subjects the Internet Gaming Disorder Scale Short Form (IGDS9-SF) for a self-rated assessment of co-occurring Internet and/or gaming addiction. [22]. The IAT (Italian version by Fioravanti and colleagues [21]) consists of 20 questions and a cut-off point of 50 as clinically meaningful for having an Internet addiction (Cronbach alpha for this measure was 0.81 in the current sample). Similarly, the IGDS9-SF [22] is a unidimensional tool including nine items reflecting all the nine criteria for the IGD, as in the DSM-5, and it assesses the severity of IGD on the basis of both online and offline gaming habits in the last twelve months. The IGDS9-SF is widely used in research on IGD and is supported by several cross-cultural psychometric studies [21] with a clinical cut-off of 21 in the Italian version [23]. In the current sample, the Cronbach alpha for this measure was 0.76.

All these measures were administered by trained child psychiatrists during multiple sessions with the patients and parents. To improve the reliability and validity of the diagnoses, after each diagnostic session, clinical data from each subject–parent pair were reviewed by the research team for the purpose of consensus, with the supervision of the senior authors (SB, GM).

After 4–6 months, a clinical follow-up was carried out in 40 subjects with an assessment procedure including the measures administered at baseline, as well as the Clinical Global Impressions-Improvement Scale (CGI-I) [11].

The study conformed to the Declaration of Helsinki. The patients and parents received detailed information on the characteristics of the assessment instruments and treatment options, and all the parents gave informed written consent. The methodology of the study was approved by the Regional Ethics Committee for Clinical Trials of Tuscany (Date 27 July 2021, Number 202/2021).

### 2.3. Statistical Analysis

Descriptive statistical procedures (means, standard deviations, ranges, and frequency distributions) were estimated to describe the study sample and assess its socio-demographic and clinical characteristics. (see Table 1); A univariate analysis with Student’s T-test for single-paired data on individual clinical groups was used to detect significant changes over time in variables with continuous distribution.

## 3. Results

### 3.1. Natural History and Gender Ratio

The study sample was predominantly composed of males, accounting for 71.25% of the total. While the mean age of the sample was 15.2 ± 1.8 years (age range 10.8–17.8 years), the onset of social withdrawal occurred at a mean age of 13.11 ± 2.0 years (age range 9.0–18.4 years), and the onset of psychiatric disorders with a first access to medical services occurred at a mean age of 11.4 ± 3.0 years (age range 6.0–16.9 years). Nineteen subjects (23.75%) of the sample had withdrawal onset corresponding to the pandemic-related lockdown for SARS-CoV-19.

### 3.2. Baseline Clinical Characteristics and Treatments

Regarding clinical severity, assessed using the CGI-S, the sample showed a marked impairment, as only 2 subjects (2.5%) of the sample scored three (mild illness); 17 subjects (21.25%) scored four (moderate disease); 38 subjects (47.5%) scored five (marked illness); 22 subjects (27.5%) scored six (severe disease); and 1 subject (1.25%) scored seven (very severe disease).

Similarly, regarding the functional impairment assessed through the CGAS ratings, most of the patients were severely impaired: 28 subjects (35% of the sample) scored between 40 and 31 (high impairment of functioning in most daily areas); 32 subjects (40%) scored between 50 and 41 (high impairment of functioning in at least some areas and some degree of impairment in most areas); and only 20 subjects (25%) scored between 51 and 60 (moderate impairment of functioning in most areas).

An exploration of the primary categorical diagnosis according to the DSM-5 criteria showed the following: 23 subjects (28.5%) presented an anxiety disorder (prevalently generalized anxiety disorder, less frequently social anxiety disorder and panic disorder); 18 subjects (22.5%) an autism spectrum disorder; 17 subjects (21.25%) a bipolar disorder; 17 subjects (21.25%) a depressive disorder; 4 subjects (5%) an obsessive compulsive disorder; only one subject (1.25%) presented post-traumatic stress disorder; and one subject (1.25%) presented an attention-deficit hyperactivity disorder and a primary categorical diagnosis.

Twenty-six subjects (32.5%) presented suicidality (intended as suicidal ideation, planning, suicide attempts, and behavior preparatory to suicide) and sixteen subjects (20%) had already made a suicide attempt.

Thirty-six out of the eighty patients (45.0%) were receiving pharmacotherapy at the baseline, while forty-five patients (56.25%) were receiving psychotherapy, and twenty-one patients (26.25%) were only monitored.

In the questionnaire analyses regarding the use of electronic devices, 25 subjects (42.37%) out of the 59 patients who completed the IGDS9-SF exceeded the cut-off, and 33 subjects (50.77%) out of the 65 patients who completed the IAT exceeded the cut-off.

### 3.3. Data from the Follow-Up

During the 4–6 months of follow-up, 24 subjects (60.0%) received intensive neuropsychiatric monitoring (see Table 2); 37 subjects (92.5%) changed their pharmacological treatment (start of new medication or dosage change); and 9 subjects (22.5%) started psychotherapy.

At follow-up, 30 subjects (75%) presented an improvement in social withdrawal, and 5 of them (12.5%) a total remission of the symptoms.

Twenty-seven subjects (67.5%) showed a global improvement, as assessed through the CGI-I (score one or two); twenty-two subjects (55%) positively changed their functioning class assessed using the CGAS scale. To assess the social withdrawal unrelated to the COVID-19 pandemic, eight subjects with pandemic-related onset of symptoms were subtracted from the follow-up sample analysis.

A significant improvement was found in the patients’ global functioning (C-GAS), from a score of 44.22 ± 6.3 at baseline to a score of 51.78 ± 8.5 at the follow-up (paired *t*-test t = −3.3; df = 31, *p* = 0.002).

Regarding anxiety, both the MASC-2 SR and PARS scores significantly improved. The MASC total t-score improved from 62.44 ± 16.2 at baseline to 58.38 ± 14.2 (paired *t*-test t = 2.3, df = 31, *p* = 0.028). The PARS total score improved from 45.75 ± 11.9 at baseline to 19.47 ± 8.1 at the follow-up (paired *t*-test t = 13.4, df = 31, *p* < 0.001). Also, self-rated depression, according to the CDI-2 total t-score, significantly improved from 67.2 ± 12.1 at baseline to 61.2 ± 13.4 at the follow-up (paired *t*-test t = 2.9, df = 30, *p* = 0.007).

Both the CBCL total and internalizing scores significantly improved. The CBCL total t-score improved from 67.63 ± 7.0 at baseline to 64.94 ± 7.0 at the follow-up, and the CBCL internalizing score improved from 73.59 ± 7.4 to 71.53 ± 7.4 (paired *t*-test t = 2.6, df = 31, *p* = 0.014 and t = 2.4, df = 31, *p* = 0.024). Among the CBCL scales, only the somatic complaints significantly improved (63.8 ± 8.5 at baseline to 61.38 ± 8.8 at the follow-up, paired *t*-test t = 2.1, df = 31, *p* = 0.045), while, of note, the scale withdrawal/depression failed to improve (79.91 ± 13.0 at baseline to 76.72 ± 12.3 at the follow-up, paired *t*-test t = 1.6, df = 31, *p* = 0.109).

The two self-reported Hikikomori measures, the Hikikomori Questionnaire (HQ-25) and the Hikikomori-related modern type of depression (TACS-22), significantly improved. The Hikikomori Questionnaire (HQ-25)’s total score improved from 57.52 ± 15.2 at baseline to 50.48 ± 19.1 at the follow-up (paired *t*-test t = 2.4, df = 30, *p* = 0.022). The Hikikomori-related modern type of depression (TACS-22)’s total score improved from 49.28 ± 10.1 at baseline to 45.69 ± 12.2 at the follow-up (paired *t*-test t = 2.1, df = 31, *p* = 0.046.

## 4. Discussion

This is one of the few studies on social isolation and Hikikomori including only adolescents, as most of the previous studies included patients with a wide age range, with adolescents and adults being included in the same samples. We can assume that adolescents with Hikikomori may present specific features, treatment needs, and changes over time, representing a specific subtype of the disorder. For example, the Fact-finding Survey on Social Withdrawal (SYPA), including subjects from 15 to 39 years, showed that a refusal to attend school was mentioned as the most frequent trigger of Hikikomori [24]. Similarly, a recent secondary analysis of the same SYPA data reported that school dropout history is an important factor associated with Hikikomori [25]. These phenomena highlight the need for a careful and specific exploration of social isolation during childhood and adolescence.

Our consecutive sample included prevalently males (71.25%), consistently with the literature. In a previous study examining a sample of Japanese population with Hikikomori, the percentage of male individuals was 65.5% [25]. Similarly, in another study of a Japanese sample aged 20–49 the prevalence of Hikikomori was four-fold in males compared to females (1.8% vs. 0.4%) [6].

Regarding the distinction between “primary” and “secondary” Hikikomori, none of our patients were categorized under the ‘primary’ form of Hikikomori [6,21], as all of our patients received a psychiatric diagnosis according to the DSM-5 criteria after a careful psychiatric assessment including a structured, diagnostic clinical interview. Consistently, in a study on sociodemographic, clinical characteristics, and possible clinical subtypes of social withdrawal in Europe [26], only one patient was characterized under the ‘primary’ form of Hikikomori, although in comorbidity with an Internet addiction disorder. The different rates of “primary” versus “secondary” forms of Hikikomori compared with Asian samples may be accounted for by differences in culture between European and Eastern countries, as Asian patients may present more frequently a “pure” form of Hikikomori [27]. In a Japanese study, the lifetime prevalence of a Hikikomori patient having a psychiatric disorder was 54%, which was higher than in the general population (29.5%) [6]. Furthermore, about 35% of Hikikomori patients met the criteria for a psychiatric disorder before the onset of social withdrawal [6]. Data concerning Western countries are not yet known, but we can assume that, beyond cultural specificities, and not alternatively, different diagnostic procedures and sampling may account for the rates of secondary Hikikomori. A careful diagnostic assessment with a structured clinical interview administered by trained child psychiatrists and including only adolescents referred to a university hospital may have increased the rate of psychiatric diagnoses.

A relevant rate of suicidality events (including suicidal ideation, planning, suicide attempts, and behaviors preparatory to suicide) was found in our sample, as up to 32.5% of subjects presented suicidality and 20% of the sample had already made a suicide attempt. These data are consistent with previous findings from Asian samples, indicating that a significantly increased suicidality risk has been reported in patients with Hikikomori [25]. Even though our sampling procedure, including patients referred to a department of Child and Adolescent Psychiatry with a high rate of mood disorders, may have inflated the suicidality of our participants, it may be hypothesized that social isolation occurring in patients with depressive or bipolar disorder may further increase the risk of suicidal behaviors. If confirmed in larger samples, this finding may underline the need of a specific assessment of suicidal risk in isolated adolescents, particularly in critical phases, such as when isolation is improving and new social contacts are becoming more frequent.

A high rate of Internet gaming disorder was found according to the IGDS9-SF questionnaire, as more than half of the patients exceeded the cut-off of the IAT questionnaire for the assessment of Internet addiction, further supporting previous findings [28,29,30] and suggesting that social withdrawal and excessive use of electronic devices may influence each other [31,32]. This finding is consistent with recent findings from an Italian sample of university students, supporting the relationship between problematic internet use and Hikikomori traits, namely, in females [33].

Regarding the short term follow-up, while parent-rated social withdrawal (according to the CBCL subscale withdrawal) did not improve, a moderate but significant improvement was found in self-rated Hikikomori symptomatology (according to the HQ-25). In a previous study [34], the CBCL subscale withdrawal resulted to correlate with the severity of Hikikomori symptomatology. There are currently no data on which measures (i.e., self-rated vs. parent-rated) are most sensitive for monitoring social withdrawal symptomatology. Our data suggest that the HQ25 may be more sensitive to detect changes in withdrawal behaviors in adolescent samples, compared to the subscale “Withdrawal” of the CBCL 6-18.

In our sample, a moderate but significant improvement was observed in the majority of patients. In a follow-up study from Spain, when treatment was more intensive (e.g., hospitalization), improvement was greater and more stable over time, even in patients with more severe psychopathology on admission [26]. In our study, after the diagnosis of social withdrawal, patients were placed in timely neuropsychiatric, pharmacological, or psychotherapeutic monitoring programs, which may have positively affected the clinical severity. Furthermore, most studies include adult patients, in which a greater difficulty in the remission of symptoms may be explained by a lesser changeability of the clinical picture, related to the longer time of illness and the greater chronicity. Thirdly, adolescent patients, although often poorly compliant, were accompanied to their visits by their family members, who may constitute an initial support network.

The main limitation of the study is the small sample size, especially for patients with a follow-up. Due to this limitation, it was not possible to make a correlation between an improvement in social withdrawal symptoms and psychopathological diagnoses. Furthermore, the patients were recruited in a third-level hospital and this type of referral may have selected the more severe patients, possibly inflating the incidence of psychiatric comorbidities. Community studies in Italian samples are warranted. The Hikikomori measures are currently still in the process of validation among Italian samples, possibly reducing the validity of the results although our methodology may have increased the reliability of the measures. Not all patients showed very good compliance when filling the questionnaires, and this did not make possible to carry out the same diagnostic protocol for the whole population, especially with regard to the questionnaires on Internet and gaming addiction. However, it must be specified that the questionnaires concerning Internet addiction or gaming addiction were frequently not completed by patients who did not present these addictive behaviors. Finally, a significant limitation is the short follow-up time (4–6 months), which made it impossible to monitor the long-term developmental trajectory of social withdrawal.

In summary, all of our patients with social withdrawal symptomatology received a clinical diagnosis according to the DSM-5 criteria and none of them were categorized under the ‘primary’ form of Hikikomori. One could therefore consider social withdrawal as a transnosographic entity, rather than a specific diagnostic category. The assessment of social withdrawal symptomatology should be closely related to the exploration of both suicidality (ideation and behavior) and the risk of Internet and gaming addiction. A short-term psychiatric follow-up revealed that, when timely diagnosed, closely monitored, and treated, a modest but significant decrease in social withdrawal symptoms was reported in 65–75% of the sample. Thus, early recognition and timely multimodal treatments may positively affect the developmental pathways of these severe patients.

Our findings suggest that there is room for further research. The study of Hikikomori in samples outside Japan is still scarce. Shared diagnostic criteria and procedures may favor a timelier identification of these patients, as well as the comparison of clinical features and developmental pathways in different countries. These studies may involve both referred and community samples, possibly presenting significant differences (particularly, the rates of primary vs. secondary forms). Gender differences should be explored for specific aspects, including psychiatric comorbidity, suicidality, association with internet gaming disorder, response to treatments, and outcome. More significantly, follow-up studies over more extended periods may help one to identify possible predictors of outcome and more specific treatment strategies, tailored for specific subgroups of patients. In this line, studies focused on young adolescents with early-onset Hikikomori may help define possible specific features of these complex patients, such as a greater changeability, but also a greater suicidal risk.

## Figures and Tables

**Table 1 children-10-01669-t001:** Baseline demographic and clinical characteristics of the sample (n = 80).

Males	57	71.25	
	Mean	SD	Age Range
Mean age (y)	15.2	1.8	10.8–17.8
Age of onset of social withdrawal (y)	13.11	2.0	9.0–18.4
Age of onset of psychiatric disorders (y)	11.4	3.0	6.0–16.9
Primary categorical diagnosis	n	%	
Anxiety disorder	23	28.5	
Autism spectrum disorder	18	22.5	
Bipolar disorder	17	21.25	
Depressive disorder	17	21.25	
Obsessive compulsive disorder	4	5	
Post-traumatic stress disorder	1	1.25	
Attention-deficit hyperactivity disorder	1	1.25	
Suicidality	26	32.5	
Suicide attempt	16	20	
Treatments			
Pharmacotherapy	36	45.0	
Psychotherapy	45	56.25	
Medical monitoring	21	26.25	

**Table 2 children-10-01669-t002:** Difference between T0 (baseline score) and T1 (end of the follow-up) in the measures of assessment (means and standard deviations). Legend: C-GAS: Children’s Global Assessment Scale; MASC: Multidimensional Scale of Anxiety for Children; CBCL: Child Behavior Checklist; IAT: Internet Addiction Test; HQ25: Hikikomori Questionnaire; TACS: Tarumi’s Modern-Type Depression Trait Scale; CDI: Children’s Depression Inventory; and PARS: Pediatric Anxiety Rating Scale. *: Statistical significance (*p* < 0.05), ** (*p* <0.01).

	Mean	Standard Deviation	T	Degrees of Freedom	*p*-Value
CGAS t0-t1	13.812	23.582	−3.313	31	** 0.002
MASC total score t0-t1	4.063	9.993	2.3	31	* 0.028
CBCL Anxiety/depression t0-t1	1.563	7.675	1.152	31	0.258
CBCL Social withdrawn t0-t1	3.188	10.929	1.65	31	0.109
CBCL Somatic complaints t0-t1	2.438	6.609	2.086	31	* 0.045
CBCL Thought problems t0-t1	2.094	9.25	1.28	31	0.210
CBCL Attention problems t0-t1	0.844	7.414	0.644	31	0.524
CBCL Rule Breaking behavior t0-t1	1.781	5.684	1.773	31	0.086
CBCL Aggressive behavior t0-t1	2.031	7.575	1.517	31	0.139
CBCL Internalizing problems t0-t1	2.063	4.918	2.372	31	* 0.024
CBCL Externalizing problems t0-t1	2.438	6.993	1.972	31	0.058
CBCL total t0-t1	2.688	5.828	2.609	31	* 0.014
IAT t0-t1	2.484	10.908	1.268	30	0.215
HQ25 t0-t1	7.032	16.226	2.413	30	* 0.022
TACS t0-t1	3.594	9.771	2.081	31	* 0.046
CDI t0-t1	5.806	11.11	2.91	30	** 0.007
PARS total t0-t1	26.281	11.055	13.448	31	** 0.000

## Data Availability

The data presented in this study are available on request from the corresponding author.

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
