# Peer review of "Hikikomori (Severe Social Withdrawal) in Italian Adolescents: Clinical Features and Follow-Up"

_children, 2023, doi:10.3390/children10101669_

Round 1

Reviewer 1 Report

General Comments:

The study on Hikikomori in Italian adolescents is certainly valuable and timely. However, there are several aspects that require substantial revisions to enhance the clarity, validity, and impact of the research. The paper should focus on refining its methodology, data presentation, and interpretation to make a more significant contribution to the field of adolescent mental health.

1. Methodology:

- Clarify the recruitment process and sampling strategy for the participant pool. Elaborate on how participants were identified and selected, including any potential biases.

- Provide more detailed information about the assessment instruments used, their psychometric properties, and their cultural adaptations for the Italian context. This will enhance the validity of the collected data.

- Clearly outline the procedure for data collection, ensuring consistency in administration and scoring of assessment tools across different researchers.

2. Data Presentation:

- Organize the presentation of results in a more structured manner. Consider using tables and figures to present key findings, making it easier for readers to comprehend complex data relationships.

- Present baseline characteristics of the participants in a comprehensive table, including demographic information, clinical diagnoses, and assessment scores.

3. Results Interpretation:

- When discussing the prevalence of psychiatric diagnoses, provide context by comparing your findings with existing literature on adolescent mental health. This will help establish the significance of your results.

- While discussing suicidality and attempted suicides, address potential confounding factors and consider discussing implications for intervention strategies or preventive measures.

4. Discussion:

- Emphasize the novelty of your study in the context of exploring Hikikomori in an Italian adolescent population. Compare your findings with international research on Hikikomori, highlighting similarities and differences, if any.

- Provide a nuanced discussion about the challenges and implications of diagnosing and distinguishing between primary and secondary forms of social withdrawal, considering cultural, social, and psychological factors.

5. Limitations and Future Directions:

- Clearly outline the limitations of your study, such as the relatively small sample size and potential biases in participant selection.

- Suggest potential avenues for future research, such as longitudinal studies that track the trajectories of Hikikomori symptoms over a more extended period and investigations into the efficacy of specific interventions tailored to adolescents.

6. Language and Clarity:

- Revise the text for clarity, conciseness, and appropriate scientific language usage. Proofread the paper for grammatical errors and typos to enhance its readability.

7. Ethical Considerations:

- Provide explicit information about the ethical approval obtained for the study, including the Institutional Review Board's (IRB) approval number and the relevant ethical guidelines followed during participant recruitment and data collection.

8. References:

- Ensure that the references are up-to-date and encompass a comprehensive review of existing literature on Hikikomori, adolescent mental health, and related concepts.

Language and Clarity:

- Revise the text for clarity, conciseness, and appropriate scientific language usage. Proofread the paper for grammatical errors and typos to enhance its readability.

Author Response

Answers to Reviewer 1 in attach file

Reviewer 2 Report

 Comments:
1. The article shows current and important content.
2. The text should be supplemented with references to publications:
- Karolina Kalita,
THE RETREATISM PROCESS OF HIKIKOMORI PEOPLE ON THE EXAMPLE OF 24 JAPANESE HIKIKOMORI CLIENTS
- Beata Szluz, "HIKIKOMORI" - THE PROBLEM OF SOCIALLY WITHDRAWN OF YOUNG PEOPLE.
3. The research methods were correctly selected. They should be described in more detail.
4. I suggest expanding the conclusions. A broader comparison of the research results with the results of other authors should be made.

-

Author Response

Answers to Reviewer 2 in attach file

Round 2

Reviewer 1 Report

The authors took the suggestions into consideration. They should review the English presentation one last time.

 Minor editing of English language required